# Recent Advances on Thermally Conductive Adhesive in Electronic Packaging: A Review

**DOI:** 10.3390/polym13193337

**Published:** 2021-09-29

**Authors:** Md. Abdul Alim, Mohd Zulkifly Abdullah, Mohd Sharizal Abdul Aziz, R. Kamarudin, Prem Gunnasegaran

**Affiliations:** 1School of Mechanical Engineering, Engineering Campus, Universiti Sains Malaysia, Nibong Tebal 14300, Penang, Malaysia; abdulalim@student.usm.my (M.A.A.); roslan_k@usm.my (R.K.); 2Department of Mechanical Engineering, College of Engineering, Universiti Tenaga Nasional, Putrajaya Campus, Jalan IKRAM-UNITEN, Kajang 43000, Selangor, Malaysia; Prem@uniten.edu.my

**Keywords:** epoxy adhesive, electronic packaging, thermal conductivity, conductive filler, thermally conductive adhesive (TCA)

## Abstract

The application of epoxy adhesive is widespread in electronic packaging. Epoxy adhesives can be integrated with various types of nanoparticles for enhancing thermal conductivity. The joints with thermally conductive adhesive (TCA) are preferred for research and advances in thermal management. Many studies have been conducted to increase the thermal conductivity of epoxy-based TCAs by conductive fillers. This paper reviews and summarizes recent advances of these available fillers in TCAs that contribute to electronic packaging. It also covers the challenges of using the filler as a nano-composite. Moreover, the review reveals a broad scope for future research, particularly on thermal management by nanoparticles and improving bonding strength in electronic packaging.

## 1. Introduction

The fast growth of the electronics industry has raised demand for epoxy-based thermally conductive adhesives to face challenges in cooling high-performance devices [1]. Traditional thermal management is not sufficient to cool high heat-producing electronic chips. As a result, high-efficiency electronic devices need proper joining material with excellent thermal performance to achieve efficiency and dependability [2]. Overheating damages materials, generates cracks and deforms the micro-level structure. It causes system failure, user health and safety problems [3] and, assembly loss in the electronics sector [4]. If heat is not removed at or above the heat generation rate, the internal temperature of the components of the device will continue to increase. Hence, it will degrade the reliability and performance. According to the US Department of Defence, the failure factor is the ratio of failure rate at any temperature over the projected temperature [5]. The increase in the failure rate of bipolar digital devices with temperature is shown in Figure 1 [6]. The graph indicates that the failure rate increases exponentially after 75 °C. Epoxy-based thermally conductive adhesive (TCA) is a potential solution for joining components and removing the extra heat generated during the device’s operation. These adhesives are used for joining chips and substrates, metal components, polymer composites, and concrete structures, which are challenging for other techniques. Epoxy adhesive polymerizes into an amorphous and highly crosslinked material. It has numerous advantages as a microstructure, including various curing methods, low curing thresholds, and a high weight resistance [7]. Epoxy adhesives have been used as a suitable substitute for traditional fasteners in various industries, including automotive, aircraft, electronics, construction, sports, and packaging [8].

The adhesive industry has changed dramatically over the last few years due to new substrates, a greater diversity of raw materials, formulations and processes, new applications, and operating conditions. Epoxy adhesives should have unique properties like- optimal cure, high-temperature service resistance and thermal cycling, and fatigue load and vibration failure resistance [9]. The epoxy adhesive is considered a composite polymer, consists of epoxy and filler materials. Fillers have recently shifted from micro to nanoscale [10]. Epoxy adhesives with nanoparticles have significantly better multifunctional properties than a conventional composite material [11].

The research interest in the conductive adhesive is rising every year. The Scopus database indicates that the number of published papers is more than 2400 in the last 10 years, related to the conductive adhesive in electronic packaging. Many review articles have also been published, with a focus on interfacial thermal resistance (ITR) [12], design and the preparation of polymer-based TIM [13], cellulose nano-fibrils [14] and, polymer/nanocarbon nano-composite [15]. However, a detailed review article comprising the fundamentals of thermally conductive adhesive (TCA), formation, and heat transfer mechanism, the summary of thermal conductivity obtained from recent articles of TCA based on ceramic, metallic, and carbon fillers is still lacking. However, the TCA demonstrated outstanding benefits and attracted interest, as mentioned by many researchers, as it is one of the most promising thermal management options in electronic packaging. Thus, the present review addresses the recent advancements in TCA formulation, reliability, and applications.

### Thermally Conductive Adhesive in Electronics Packaging

The rapid advancement of microelectronic technology has resulted in the progressive transformation of electronic components from isolated to highly integrated. It produces a lot of heat while they are functioning [16]. The materials and structures used to attach the semiconductor chip to other electronic components involve the sector of electronic packaging. Conductive adhesives represent a significant advancement in electronic packaging for advanced devices, where effective heat dissipation and enough electrical conductivity are critical [17]. The TCA joint is a crucial choice for the thermal management of the device. The advances in TCA are essential because of future demands for inexpensive and handy electronic devices [18]. The adhesive applications have become more important due to their remarkable versatility and unparalleled compatibility. The electronic industry has indeed developed, especially in consumer electronics, which depends on conductive adhesives. It would not be possible to identify the most modern electronic devices without conductive adhesives. It is an inherently clean and convenient solution for solders in high-density electrical connections [19]. TCAs are applied in particular tasks in electronics industries like die attachment process in LED packaging, PCB fabrication, advanced thermally conductive composites production, chip-scale packages, and power semiconductors. The sector-wise applications of TCAs are shown in Figure 2.

The need for effective heat conduction within electronic equipment has been highlighted more with the development of 5G mobile networks. As a result, thermal interface materials are essential in today’s modern electronics to ensure efficiency and reliability. Polymer composites combine the benefits of a polymer matrix and a thermally conductive filler. To tackle the problem of heat conduction, thermally conductive polymer composites are widely used [17].

The lead (Pb) containing solder alloys were commonly used as interconnect materials in most electronic packaging areas. Many of these items have a short life span, and millions of those that end up in landfills are contaminated with lead. It is a big challenge to recycle lead-containing electronics goods [20]. Few developed countries have already banned the manufacturing and importing of all lead (Pb) containing electronics. The use of lead-free solders is gradually dictated because of environmental problems and health concerns. Thousands of tons of lead are used every year to produce electronic items [21]. To avoid Pb/Sn soldering attempts, the findings of major electronics manufacturers have begun using TCA as alternatives. The chemical composition of TCA includes both inorganic (metal or ceramic) particles and an organic (polymer) compound [22].

## 2. Formulation of Thermally Conductive Adhesive and Heat Transfer Mechanism

A polymer matrix and thermally conductive filler make thermally conductive adhesives that are formed as composites. Nanoparticles are having a diameter of less than 100 nm exhibit unique physical and mechanical properties. These fillers have a large specific surface area, high surface energy, and interfacial area with the polymeric matrix [23]. Thermal conductivity of epoxy is inadequate to remove heat produced from electronic devices. Dispersing nanoparticles inside an epoxy adhesive matrix can dramatically improve the adhesive’s thermal properties [10] and the performance of adhesively bonded joints [24]. The high thermal conductive solid particles like—graphene [25], carbon nanotubes [26], carbon fiber, carbon black are the most common carbon-based fillers. Metallic (gold, copper, silver, aluminium) [27], and ceramic (aluminium nitride, boron nitride, silicon carbide, aluminium oxide) [28] fillers have been blended with matrix resin to increase the thermal conductivity. Carbon-based and metallic filler adhesive has received significant attention because of its excellent thermal conductivity and specific surface area [29]. These conductive filler particles can alter polymer matrix isolation properties by making them very heat conductive. The thermally conductive adhesive is a composite material, and it follows the same formulation process of nano-composite [30]. Most common formulation process starts with the filler’s dispersion with organic solvents. An example of graphene nanoparticle (GNP)-based nano-composite [31] is shown in Figure 3. The process until the application, i.e., (a) to (h), is maintained for all thermally conductive adhesive according to the filler type. Bubbles and voids reduce the bonding quality of the adhesive [32]. To avoid the bubble and void in the adhesive joint the vacuum degassing is important in thermally conductive adhesive production process.

The non-conductive properties can be transformed into conductive properties with the addition of fine conductive filler particles in polymer matrices, resulting in a continuous thermal conductive network [33]. The thermal interface material (TIM) required high thermal conductivity. The thermal conductivity of polymers ranged from 0.1 to 0.5 Wm^−1^ K^−1^ [34]. The simple form of polymer matrix is insufficient to meet the thermal conductivity requirement. Conductive filler particles increase heat conductivity while conserving polymer matrix characteristics [35]. Before integrating the epoxy with the filler, it is essential to know the thermal conductivity of the filler. A list of widely used conductive fillers is presented in Table 1.

Electrons and phonons are responsible to heat transfer in metal. Due to the lack of free electrons in non-metals, the phonon is responsible for heat transmission. When a polymer surface comes into contact with a heat source, heat is transferred via vibrations to the nearest atom, then to the next, and so on [41]. The formulation of thermally conductive adhesives consists of a polymer matrix, filler, and specific additives that are uniform and isotropic at the macro scale.

The thermal conductivity of the polymer matrix is less than 0.5 Wm^−1^ K^−1^. Thermal conductivity is improved by incorporating a highly thermally conductive filler into the insulating polymer matrix. The filler has a significant influence on the thermal conductivity of polymer composites. In the case of the filler network, it can speed heat transmission along the continuous and integrated filler network with less phonon scattering. Significantly, optimizing the filler structure (including size and aspect ratio) and dispersion can improve thermal conductivity by arranging the filler in a thermally conductive network within the polymer matrix [42]. Therefore, heat transmission across TCA occurs when filler particles establish a conductive path. A general heat transfer mechanism of TCA is shown in Figure 4.

The basic principle of heat conductance through a thermally conductive adhesive layer dx (Figure 4), also known as Fourier’s law [43], states that the rate of heat transfer through a material is proportional to the negative gradient in the temperature and to the area at right angles to that gradient through which the heat flows and can be written as follows: (1)$$\frac{dQ}{dt}=-\lambda A\frac{dT}{dx}$$
where Q is the quantity of heat energy (J), *t* is time (s), *λ* is a thermal conductivity (Wm^−1^ K^−1^), *dT*/*dx* is the temperature gradient in the heat flow direction (K/m), *x* is the distance along the direction of heat flow (m), *A* is the area of the cross-section (m^2^). The thermal conductivity is given by: (2)λ=⅓(CeVeLe+CphVphLph)=λe+λph
where *Ce* and *Cph* are the heat capacities per unit volume (J/m^3^ K) of electrons and phonons, respectively, *Ve* and *Vph* are their root-mean-square velocities and *Le*, *Lph* are their mean free paths. The thermal conductivity of electron type *λ**e* is dominant for metals, and one can roughly assume that: (3)λ≅λe

In such a case, the ratio of thermal conductivity *λ**e* and electrical conductivity *σ**e*, according to Wiedemann–Franz’s law, can be expressed as:(4)$$\frac{\sigma e}{\lambda e}=LT\;$$
where *T* is temperature (K) and *L* is the Lorenz constant, the theoretical value of which is 2.44 × 10^–8^ W·Ω/K^2^. The transport of heat in nonmetals occurs mainly by phonons. For insulators, thermal conductivity increases linearly with temperature, and the efficiency of phonon heat conduction is size dependent. In nanostructures, the thermal conductivity of a phonon type *λ**ph* may be drastically different than in macrostructures. When the size of a nanostructure approaches or exceeds the mean-free path of a phonon, phonons collide with the border more frequently than in bulk materials. This extra collision mechanism increases the resistance to heat transfer, hence lowering the effective thermal conductivity of thin films, wires, nanotubes, and other nanoparticles [30].

## 3. Adhesives with Improved Conductivity

Epoxy is inferior in heat conductivity. However, research in electronic packaging is improving the heat conductivity of nanoelectronics epoxy adhesion. The conductivity range of TCAs is reported as 1–30 Wm^−1^ K^−1^ for sufficient heat transfer [44]. The thermal conductivity of epoxy has been increased using ceramic [45], metallic [46], and carbon-based [47] conductive fillers to increase the thermal conductivity of the epoxy, which are described below.

### 3.1. Ceramic-Based Fillers

Chip power density is a significant factor in the performance of electronic devices. TCAs minimize the thermal resistance between the electronic devices’ cooling medium and the outside environment. A ceramic substance is used to insulate to prevent electrical shorts [48] and enhance electronic equipment’s effectiveness [49]. Highly thermally conductive ceramic fillers have been added in epoxy adhesive systems, such as boron nitride (BN), aluminium nitride (AlN), alumina (Al_2_O_3_), and silicon carbide (SiC) particles. Figure 5 shows various types of thermally conductive ceramic used in TCAs.

#### 3.1.1. Boron Nitride

Boron nitride (BN) has the same number of boron (B) and nitrogen (N) atoms and is isoelectronic to carbon structures [50]. These particles have received considerable interest because of their excellent characteristics in all aspects. It is also referred to as “white graphene” because of its honeycomb composition. It has an insulating property with a bandgap of 5.2 eV and possesses considerable thermal conductivity (experimentally determined 360 Wm^−1^ K^−1^ and theoretically 2000 Wm^−1^ K^−1^ [51]. The findings are outstanding since the dielectric characteristics and the anti-oxidation resistance were found to be excellent. Electronic devices require better thermal management to match the rising demand for BN and epoxy mixed composites. The hexagonal form of BN (h-BN) [52] is comparable to graphite, the cubic form (c-BN) is like diamond [53], and the amorphous form (a-BN) is similar to amorphous carbon [54]. Several BN/epoxy composites have been created. However, a hybrid BN solution was unable to be found since the interfacial thermal resistance was too high [55]. BN nanoplates filler affects thermal conductivity of the composite with the increase in filler concentrations. It is found that the 2D-BN nanoplates filler at 14 wt% with silicone can reduce the mechanical strength, but the thermal conductivity is 30% better than the without filler sample. The reduction in the mechanical strength can be due to agglomeration of BN nanoparticles, whereas the increase in the thermal conductivity is due to the acceleration of phonons transmission by the BN nanoplates [6,56].

C. Xiaoa et al. [57] used hollow boron nitride microbeads (BNMB) at 65.6 vol% for improving thermal conductivity of epoxy resin. The result shows that the maximum thermal conductivity reached 17.61 W/m^−1^ K^−1^ in-plane direction and 5.08 W/m^−1^ K^−1^ for out-plane direction of BN.

#### 3.1.2. Alumina

Alumina (Al_2_O_3_) is used in electronics packaging as a low-cost filler. It has high thermal conductivity (30 Wm^−1^ K^−1^) and electrical insulation properties. Because of these unique properties become an essential filer in the thermal management of electronic devices [58]. It has been used in light-emitting diode (LED) packaging to avoid voltage drop, short circuits, or noise reduction. Alumina nanoparticles can significantly improve the mechanical properties of epoxy adhesives and interfacial wettability with an aluminum substrate [59]. A mixture of Bisphenol-F epoxy resin with 80% of 30 μm and 20% of 5 μm spherical Alumina (S-Al_2_O_3_) particle can increase thermal conductivity up to 1.364 Wm^−1^ K^−1^ [60].

A study by Mai et al. [61] reported that epoxy adhesive with Alumina contained hybrid filler can improve the thermal conductivity of the composite materials. Another experimental and simulation work was performed using Al_2_O_3_ and BN in the epoxy polymer (EP) matrix. The thermal conductivity of the BN/Al_2_O_3_/EP composite is higher than that of the Al_2_O_3_/EP composite. The thermal conductivity increases from 2.77 Wm^−1^ K^−1^ to 3.35 Wm^−1^ K^−1^ for the composite without BN loading (shown in Figure 6). Graphene coated alumina was used as a thermal conductivity enhancer in epoxy composite and was found to be a potential filler [62].

#### 3.1.3. Aluminum Nitride

Aluminum nitride (AlN) particles are a promising filler material in electronics packaging. The size of the AlN filler determines the characteristics of AlN-filled epoxy composites [63]. The thermal conductivity, based on AlN’s particle size, ranges from 200 to 320 Wm^−1^ K^−1^ at room temperature. The maximum amount of particle in epoxy composite does not ensure the highest thermal conductivity. Research result shows that 1 wt% nano-AlN sample has superior electrical insulation and thermal conductivity among the pure epoxy, 0.5 wt%, 1 wt% and 2 wt% of AlN/epoxy solution [64]. However, magnetic-aligned AlN/epoxy composite at low filler content can effectively generate thermal transport channels and enhance thermal conductivity from 0.915 Wm^−1^ K^−1^ to 1.754 Wm^−1^ K^−1^.

The thermal conductivity of adhesive is mainly determined by the filler heat transfer capacity, density of thermal network, as well interfacial thermal resistance. Thus, the formation of effective thermal flow 3-D percolating network through synergistic effect in matrix is a crucial criterion, dominating the thermal conductivity. Yuan et al. [65] worked on different sized (5 µm, 2 µm and, 50 nm) AlN- with graphite and graphene oxide(GO) as a hybrid filler to observe the improvement of thermal conductivity of composite. The results demonstrate that large particles of AlN with epoxy are more heat conductive than small particles. Similarly, GO can improve the thermal conductivity of epoxy resin more effectively than natural graphite. In the case of a single filler, adding 70 wt% 5 µm-AlN particles to the epoxy resulted in the maximum conductivity which is 10.8 times that of pure epoxy (shown in Figure 7).

The heat conductivity of epoxy adhesive containing GO is more effective than natural graphite. For single fillers, 10.8 times higher than pure epoxy is the most excellent conductivity AlN’s thermal expansion coefficient (CTE) is low. It is relatively low-cost, non-toxic, and can provide a stable crystalline structure [66].

#### 3.1.4. Silicon Carbide

The researchers are interested in the Silicon Carbide Filler (SiC) as it has greater hardness and strength, good resistance to corrosion and oxidation. SiC has high intrinsic thermal conductivity (490 Wm^−1^ K^−1^), more than three times higher than silicon and 10 times higher than gallium arsenide and sapphire [67]. Nanowire form of SiC is a primarily familiar shape for dispersing with epoxy. Most of the recent works are focused on hybridization with SiC nanowire. According to Dianyu Shen et al. [68], the thermal conductivity of 3 wt% SiC nanowire with epoxy is 0.449 Wm^−1^ K^−1^, which is 1.06 times higher than plain epoxy. Another experimental output found thermal conductivity 0.43 Wm^−1^ K^−1^ at 3.91 vol% of SiC nanowire epoxy composite [69]. BN and SiC hybrid filler with vertical alignment was examined, and thermal conductivity enhancement was found [70,71]. Carbon fibre (CF) was also used with SiC to improve the heat transfer properties and saw satisfactory improvement [72]. A summary of research output from recently published works on ceramic fillers are given in Table 2.

### 3.2. Metallic Fillers

Metal nanoparticles (NPs) have recently gained popularity due to their unique properties such as low melting temperature and high diffusion coefficient [91]. Research has shown that epoxy conductive adhesives enhance their heat conductivity when integrated with metal fillers. Sphere, fibre, granules, or flakes may be metallic particles. The optimal shape can be such that the filler levels of the surrounding metallic parts are the lowest crucial, well interacted, and the best matrix-resin adhesion can be found. These criteria are suitable for metallic flakes because of their high aspect ratio [92]. However, polymers loaded with metal are required with the required reinforcement. The amount of filler may be readily changed to adjust thermal conductivity. The material for heat sinking requires a low coefficient of thermal expansion (CTE) since the semiconductor chips have low CTE content. Thus, it has good thermal conductivity, and low CTE needed for thermally conductive material. Cu is usually not utilized in TCAs. Because Cu is thermally good yet has a high CTE content, Figure 8 shows some metal fillers used to manufacture TCAs.

Copper nanoparticle paste is used for bonding chips and different metallic substrates at low temperatures. The result shows that good bonds are mainly achieved with Cu and Au surfaces [93]. It provided good thermal conductivity and shear strength (20 MPa) when sintered at 350 °C [94]. The thermal conductivity of the composite was three times higher than pure epoxy at 15 wt% of MWCNT, and 34 wt% of copper nanoparticles [95], TiO_2_ coated copper nanowires 0.2–1.12 Wm^−1^ K^−1^ [96] and 2.59 Wm^−1^ K^−1^ [97] were found for a length of up to ~40 μm and diameter of ~20 nm.

Silver (Ag) particles exhibit maximum thermal conductivity among functional metallic fillers. Silver nanoparticles have a vast application in catalysis, conduction, antibacterial and electrical devices [98]. Silver nanoparticles have different forms and sizes to obtain a high heat conductivity and decrease total product costs. Silver’s thermal conductivity is 450 Wm^−1^ K^−1^, although its conductivity improvement with an epoxy adhesive is lower. Silver/epoxy self-constructed nano-structured networks led to two times better thermal conductivity than plain epoxy [99]. Ag particle at 25.1 vol% of Ag- decorated BN nanosheets provides 3.06 Wm^−1^ K^−1^ [100] and silver-decorated MWCNT/epoxy adhesive 0.88 Wm^−1^ K^−1^ [101]. Ni decorated MWCNTs as a reinforcement provides 0.30 Wm^−1^ K^−1^ [102]. The best part of Ag nano particle is that it improves the thermal conductivity and the shear strength of the adhesive joint [103], which is essential for the electronic packaging reliability [104]. Additionally, some composites of silver-coated copper, silver-coated reduced, graphene oxide nanoparticles, and graphene nanosheets embedded in the epoxy resin provide good thermal and mechanical properties [105]. Another research observed the thermal conductivity of epoxy adhesive filled by eight different filler- ZnO powders, BN powders, Al_2_O_3_ powders, graphite flake, Al powders, Cu powders, diamond powders and Ag powder. The results indicate that each sample is capable of significantly increasing the thermal conductivity of the epoxy resin. The highest thermal conductivity (1.68 Wm^−1^ K^−1^) was obtained in the graphite-epoxy adhesive at 44.3 wt%. Meanwhile, the layer-shaped filler and sharp corner-shaped fillers are preferable for improving the thermal conductivity of epoxy resin [106]. The morphologies of the eight fillers samples after spray-gold treatment were observed by SEM images (shown in Figure 9).

### 3.3. Carbon-Based Fillers

Recently, the most promising filler material has been carbon-based filler. Carbon-based fillers have excellent thermal conductivity, corrosion resistance, and low thermal expansion coefficient. Carbon–carbon composites consist of a carbon matrix, which exhibits great thermal conductivity and is, therefore, suitable for use as a heat sink [107]. However, due to the high cost of carbon-carbon composites, industrial applications are difficult. As a result, many researchers are focusing their efforts on developing low-cost thermally conductive adhesives based on carbon-based fillers. Thermal conductivity is high due its small surface area. The commonly carbon-based fillers are graphite, carbon nanotubes, reduced graphene, graphene oxide, carbon black, and carbon fiber. Graphite-based nano-composites are applied in precision flexible electronic devices [108]. Carbon nanotube-based adhesive is used to join high-performance composite parts like wing and fuselage components [109]. Graphene’s unique qualities continue to astound researchers. It has opened the way for applications including light-emitting diodes (LEDs), biosensors, batteries, 3D bio-printing, conductive inks, and touchscreen systems [110]. Graphene laminated conductive coating on plastics bodies are used in electronics to improve the thermal management system [111].

Fiber form is preferable to the particle to be more dispersible and effective in a thermal conductive composite. The [112] fiber is made from natural cellulose, synthetic polyacrylonitrile (PAN), and pitch and is carbonized or graphitized at high temperatures to remove other chemical elements and create fiber structures (shown in Figure 10) [113].

#### 3.3.1. Carbon Nanotubes

Carbon nanotubes is the one-dimensional allotropes of carbon. Ambitious research interests on carbon nanotube are found due to their exceptional material properties. Carbon nanotubes (CNTs) and their composites exhibit extraordinary physical, chemical, and electrical capabilities, opening up exciting potential for nanometer-scale electronic applications [114]. Silicon complementary metal oxide semiconductor (CMOS) device scaling is expected to end soon, but alternative technologies capable of sustaining computing power and energy efficiency have yet to be found. Carbon nanotube-based electronics have been shown to be one of the most promising possibilities [115]. Future wireless communication technologies will require integrated radiofrequency devices operating at frequencies above 90 GHz. Carbon nanotube field-effect transistors may be suitable [116]. These have been used in different practical applications. Single-wall nanotubes (SWCNTs) have a thermal conductivity of 6000 Wm^−1^ K^−1^, while MWCNTs conduct up to 3000 Wm^−1^ K^−1^ [117]. CNTs have a high aspect ratio that can considerably contribute to the total thermal conductivity. CNT composites can be utilized in the production of miniatures for thermal control by using thermally conductive polymers. [118]. CNTs also have problems due to low dispersion capabilities in the matrix resin. CNTs have a diameter in a nanoscale range and a considerable attraction of van der Waals. CNTs generally form aggregates. In particular, greater concentration of CNTs fillers dispersion is exceedingly challenging. Several types of research have been carried out for practical dispersal challenges. The most common methods of dispersal are sonication, wrappings with polymer chains, alignment using electric and magnetic fields, and decorating CNT surfaces with metallic fillers. CNT can be dispersed by functionalization and chemical modifications also. The alignment of CNTs with high-intensity magnetic fields is another effective technique to improve the thermal conductivity of epoxy/CNT [119].

CNTs are regarded as suitable bridge materials for improving the electrical, mechanical, and thermal properties of polymer nanocomposites incorporating different fillers [120].

#### 3.3.2. Carbon Fiber

Carbon fiber is an alternative to carbon nanotube due to its low cost and easy accessibility. However, it has gained less attention than carbon nanotubes because it has lower mechanical properties, with a larger diameter and higher density. Research works on carbon fiber are more cost-effective than CNTs [121]. The research findings can be used in research related to CNTs. The CTE value of polymers usually is high. The carbon fiber used in composite designs must have a minimal or negative CTE since it is crucial for dimensional stability [122]. The overall thermal conductivity of the composite is impacted by the aspect ratio and thermal conductivity of carbon fiber. The thickness of fiber and the curing pressure have also affected the thermal conductivity [123]. An experiment on enhancing thermal conductivity of carbon fiber/cyanate by developing an interfacial path for was carried out by X. Zheng et al. [124]. The thermal conductivity was found to be 0.97 Wm^−1^ K^−1^, which is 1.06 times higher than original carbon fiber/cyanate composite.

#### 3.3.3. Graphene

Andre Geim and Kostya Novoselov received the 2010 Nobel Prize in Physics “for pioneering work on the two-dimensional material graphene [125]. Because of their unique mechanical and thermal characteristics and high thermal conductivity, graphene nanosheets have a great potential to be employed in polymers as reinforcement. Graphene possesses 130 GPa tensile strength, a Young’s 1 TPa tensile modulus, and a thermal conductivity of 600–5000 Wm^−1^ K^−1^ [126]. Graphene can integrate a large specific surface area (SSA) with a high conductivity to protect electromagnetic induction (EMI) waves during transmission, and graphene materials are also broadly utilized in electrostatic discharge (ESD) and shielding [127]. A study reported by Yang et al. [128], demonstrated that a scalable highly conductive thermal films can be made by reducing GO films by facile chemical reaction using hydroiodic acid/acetic acid vapors at low temperature for application as heat dissipation. Graphene is utilized to increase the thermal conductivity of epoxy glue with minor adjustment in the heat transfer rate as a potential conduction filler. Chen et al. [129] examined the influence of filler loading on composite thermal conductivity. The findings demonstrated that graphene in any form is more successful at increasing the thermal conductivity of composites than carbon nanotubes and other carbon-based nanofillers (shown in Figure 11).

Many researchers have conducted different graphene treatments to increase nano-composite heat conductivity. Epoxy with hybrid graphene oxide (GO) and CNT filler system are the most popular techniques that make TIM a combination with GO covering and epoxy/graphene flakes. Also, the thermal conductivity value depends on the filler [130], alignment [131], interaction between filler and matrix [132], graphene layer number and particle size [133]. A summary of research output from recently published works on carbon-based fillers are given in Table 3.

## 4. Economic Perspective of TCA

Demand for conductive adhesives is increasing globally, particularly in Asia Pacific and North America. This expansion is fueled by profitable technological businesses. Asia Pacific currently leads the world in conductive adhesive sales. China and India have the largest manufacturing and consumption markets, thereby contributing the most to the worldwide conductive adhesive market. North America is the second largest consumer of conductive adhesive due to big industrial and healthcare industries. Europe, the Middle East and Africa are predicted to rise rapidly due to increased adoption and industrialization [144]. COVID-19 reduced market growth significantly last year and is expected to continue through 2021. Huge demand for the TCA market can be seen from massive manufacturing countries affected by coronavirus [145]. According to a report published in August 2020 by marketstudyreport.com (accessed on 22 September 2021), the global conductive adhesive market is estimated to increase at a compound annual growth rate (CAGR) of 4.6% from 2020 to 2025, from USD 227.3 million in 2019 [146].

The competition among the manufacturer is increasing with expansion of adhesive market. Research and development of TCA is also focused on environmental issues as well as commercialization of product. For a new green process to be commercially feasible, the techno-economical factor plays a significant role [147]. Highly compatible polymer-based conductive adhesive provides an environmentally friendly lead-free alternative to lead-based components. Nanofillers such as graphene, CNT and quantum dots are studied widely to be used to enhance the thermal conductivity of thermally conductive adhesives. However, their bulk scale utilization is limited due to the high cost and environmental concerns. Many studies have been conducted to synthesize nanofillers with low cost, facile and environmentally friendly methods. For instance, bulk-scale graphene can be synthesized from GO by a facile chemical reaction using natural substances [148]. Few-layer graphene can also be produced using electrochemical intercalation process [149]. Low-cost and wearable electronics will require conductive adhesives. The CNT market is projected to deal these demands. The ceramic filler-based adhesive has important uses in the electronic where the electrical insulation is very important. The metallic filler-based adhesive is popular for its low cost. These amazing versatility and compatibility of the adhesive’s early and wide-ranging applications has received much more attention. Since conductive adhesives are essential to the identity of most modern electronic items, they are essential to the electronic industry.

## 5. Challenges and Research Potential

Several factors influence the thermally conductive adhesive (TCA). Lead (Pb) was most commonly used in the TCA industry, however it is now restricted due to health and environmental concerns. To overcome the difficulties, researchers concentrated on incorporating nanoparticles into epoxy to enhance the adhesive performance. Due to the increasing downsizing and power density rise in electronic devices, high-performance TCAs are essential. Carbon nanotubes (CNT) and graphene have excellent mechanical and thermal characteristics, making them appropriate for use as TCA fillers. While these research results are satisfactory, commercialization is challenging due to large-scale manufacture and high cost. Next-generation TCAs should be made with lower manufacturing cost and facile methods. Using low-cost substrate materials and high-thermally conductive filler materials should be prioritized. Thermal management of electronic devices is becoming easy by developing the TCAs, but we still face challenges to produce the TCAs The common factors that affect a TCA’s performance are summarized in Figure 12.

Based on this extensive review of the recent research, it is found that the shape of the filler is a critical factor but often overlooked in the thermal conductivity improvement of TCAs. Other big challenges to progress in the TCA sector include developing low-dimensional materials with a high aspect ratio, dispersing them into the matrix, achieving high thermal conductivity while being electrically insulating, and developing a novel heat conduction network. Undoubtedly, high thermal conductivity for a TCA is one of the biggest challenges, but the following research potentials are for improving thermal conductivity of TCAs:(1)The improvement of nanomaterial preparation techniques and process parameters can contribute to the development of an efficient three-dimensional thermal conductive network in matrix.(2)Surface modification seems to be an effective method of reducing the thermal resistance at the interface; however, it leads to reduction in the low-dimensional materials’ intrinsic thermal conductivity.(3)Metallic and carbon based (graphene and carbon nanotubes) fillers have high thermal conductivity but possess high electron mobility. The ceramic fillers (BN, AlN, SiC, and Al_2_O_3_) are highly thermally conductive but electrically insulated. Therefore, the hybridization of the filler can be a new research direction.(4)The wonder material graphene oxide (GO) appears to be a potential choice because of its solution processability and controllable deposition on the substrate. GO has been used on various substrates, but there is still room for development in terms of adhesion and heat transmission.

## 6. Conclusions

This study reviewed recent studies on thermally conductive adhesives (TCA). TCA-based solutions are cost-effective and eco-friendly. They have demonstrated outstanding benefits and attracted interest from many researchers, as this is one of the most promising options of thermal management in electronic packaging. The review addresses the recent advancements in TCA formulation, reliability, and applications. Thermally conductive adhesives with superior performance and long life have been developed in direct die attachment, flip-chip, and surface-mount connect applications. TCAs are now being investigated and designed with outstanding performance and reliability for high-power devices. Various conductive fillers, such as carbon, metal, and ceramics, are used to improve the thermal conductivity of epoxy adhesives. Often, a greater wt% of fillers might lead to higher thermal conductivity. However, the mechanical characteristics of the adhesive deflect more in higher wt% of filler. Considering the state of research and development of TCAs, dynamic simulation of thermal conduction of epoxy-based conductive adhesive should be carried out using mathematics and computer software.

## Figures and Tables

**Figure 1 polymers-13-03337-f001:**
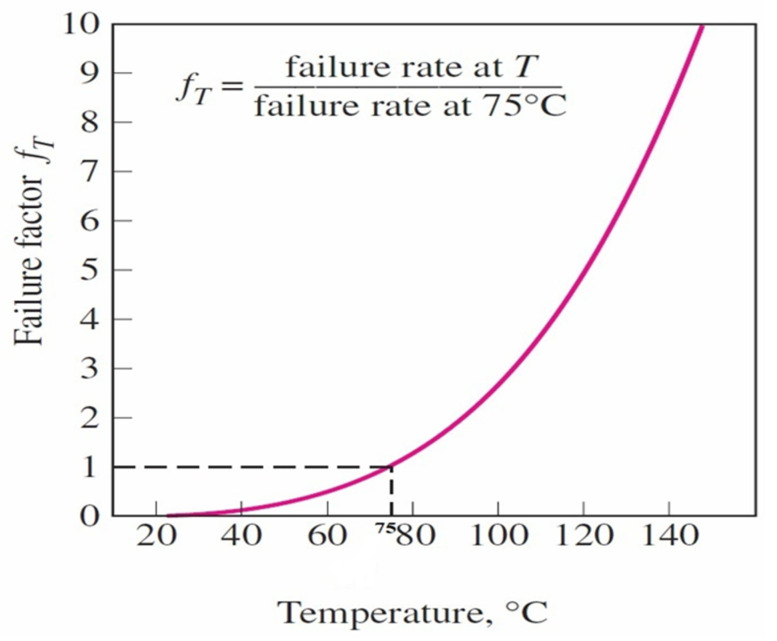
Graph of failure factor vs. temperature of a device [6].

**Figure 2 polymers-13-03337-f002:**
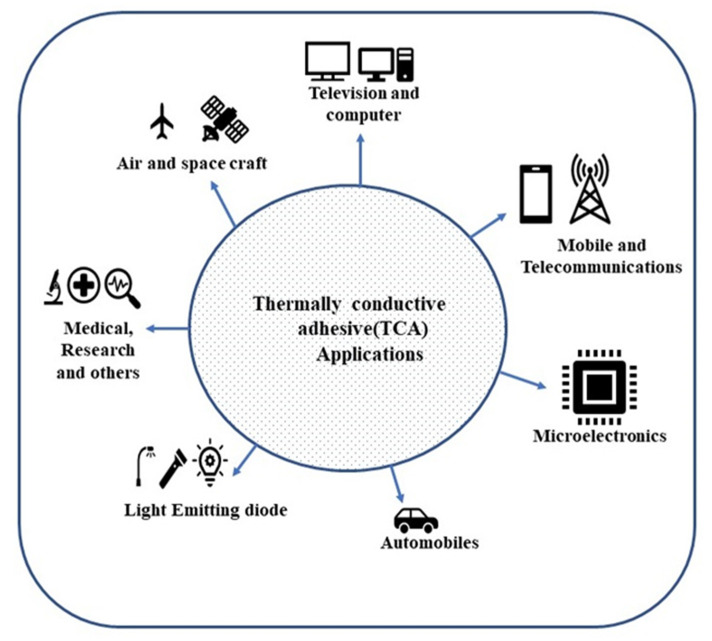
Application of TCAs in different sectors.

**Figure 3 polymers-13-03337-f003:**
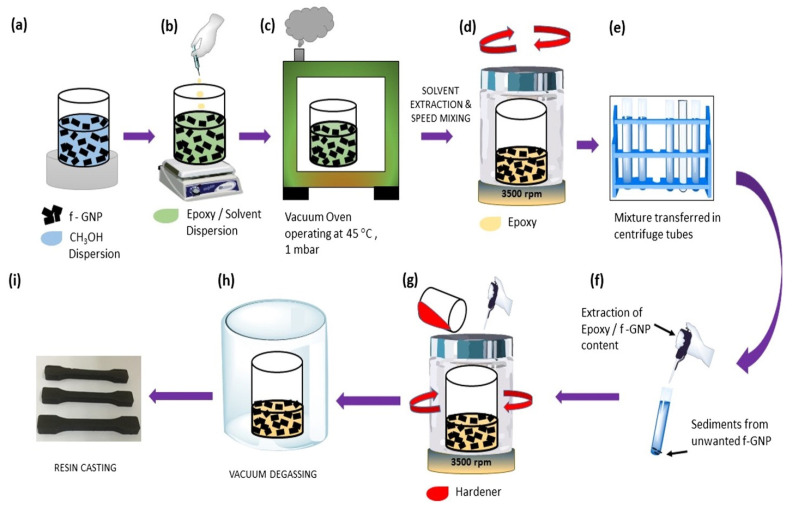
Schematic of the dispersion (**a**) to application (**i**) process of conductive filler in nano-composite [31].

**Figure 4 polymers-13-03337-f004:**
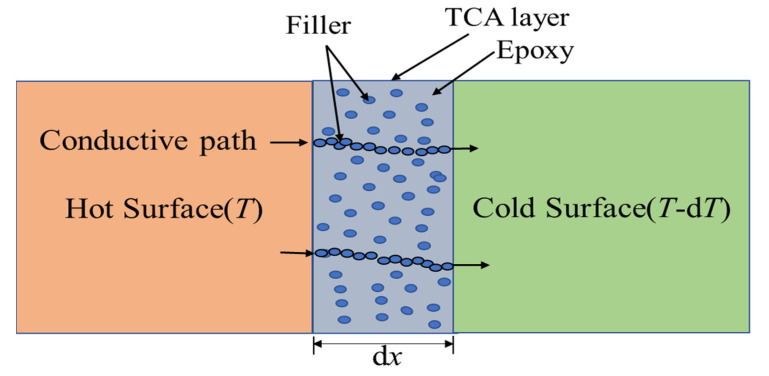
Heat conduction mechanism of TCA.

**Figure 5 polymers-13-03337-f005:**
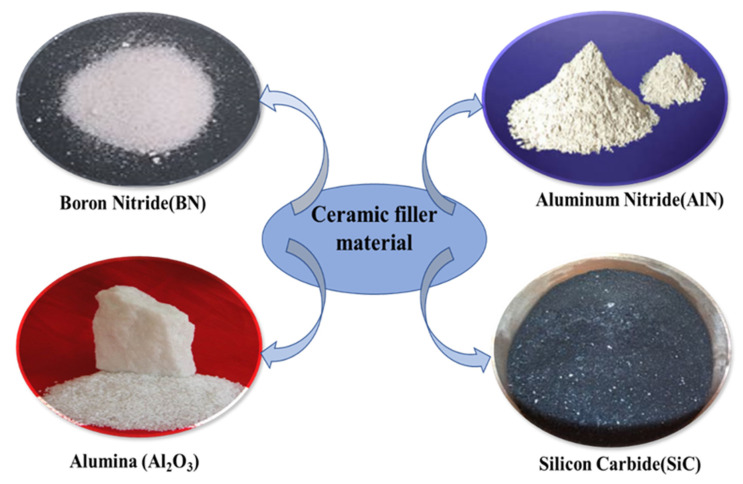
Commonly used thermally conductive ceramics filler.

**Figure 6 polymers-13-03337-f006:**
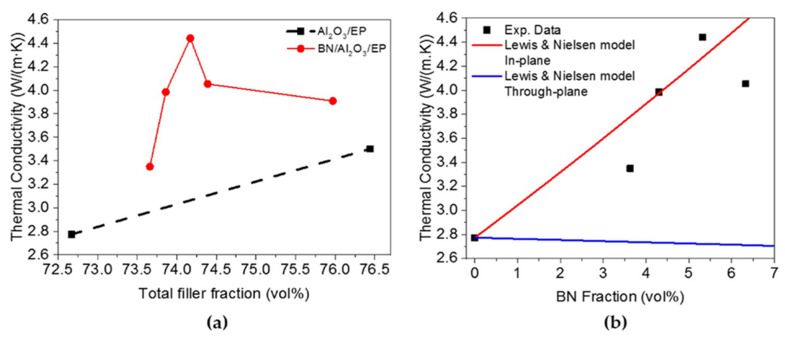
(**a**) Thermal conductivity vs. total filler fraction graph (**b**) Experimental and simulated data comparison graph [61].

**Figure 7 polymers-13-03337-f007:**
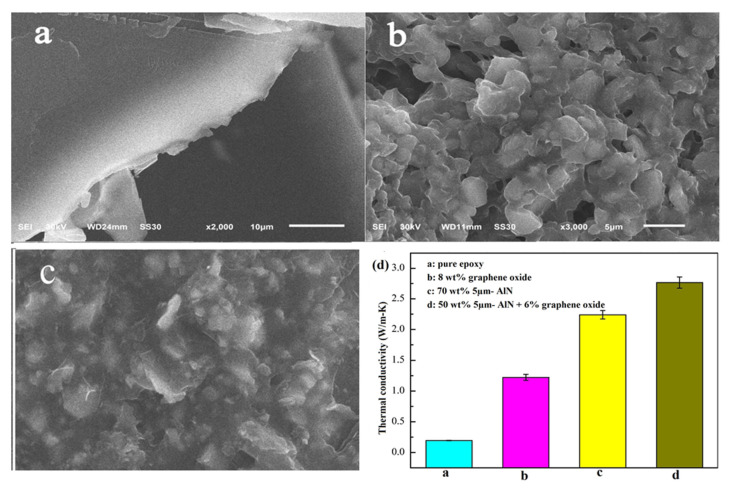
SEM images of epoxy adhesives containing (**a**) only epoxy resin, (**b**) epoxy with AlN particles, (**c**) epoxy with AlN and GO hybrid filler and (**d**) Thermal conductivity of different test samples [65].

**Figure 8 polymers-13-03337-f008:**
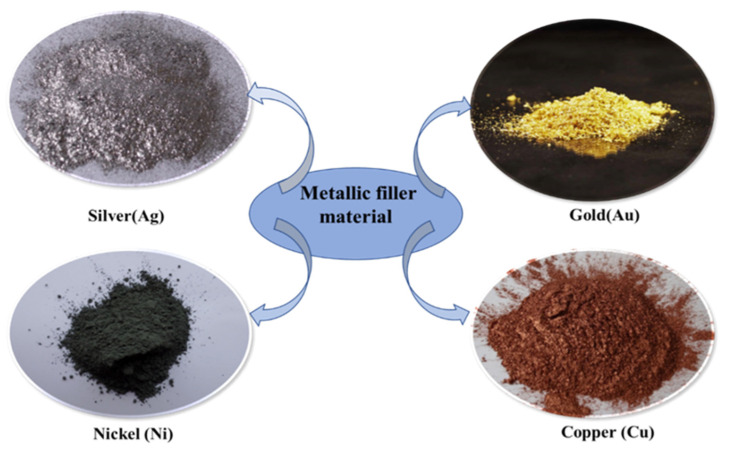
Commonly used thermally conductive metallic filler.

**Figure 9 polymers-13-03337-f009:**
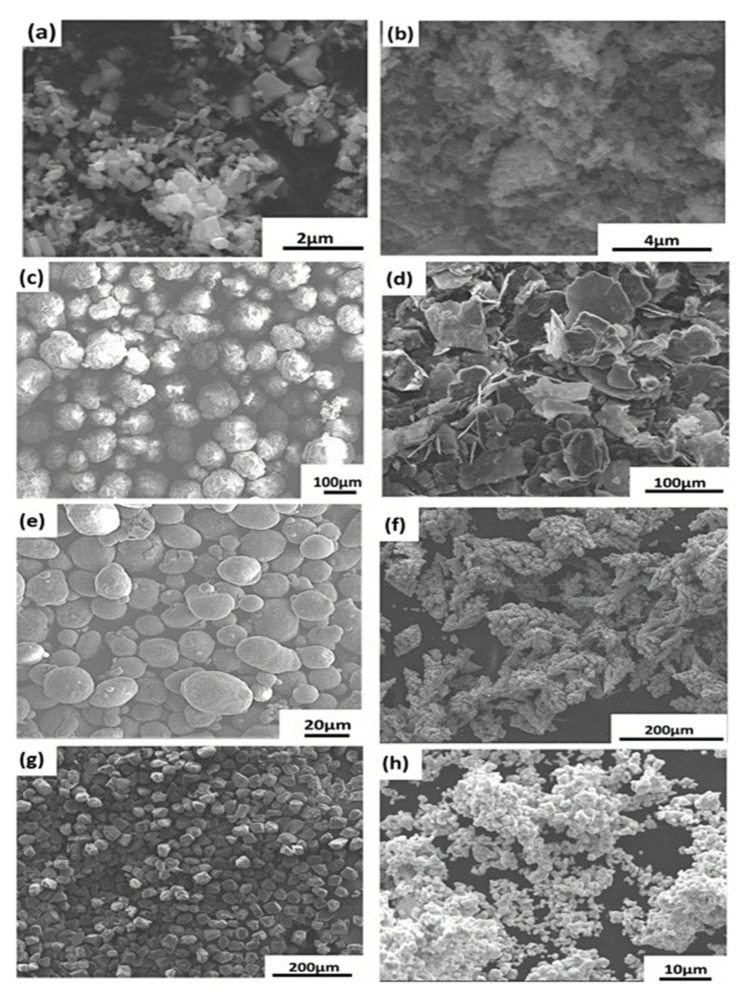
SEM images of fillers sample as powder form. (**a**) ZnO, (**b**) BN, (**c**) Al_2_O_3,_ (**d**) graphite flake, (**e**) Al (**f**) Cu (**g**) diamond and, (**h**) Ag powders [106].

**Figure 10 polymers-13-03337-f010:**
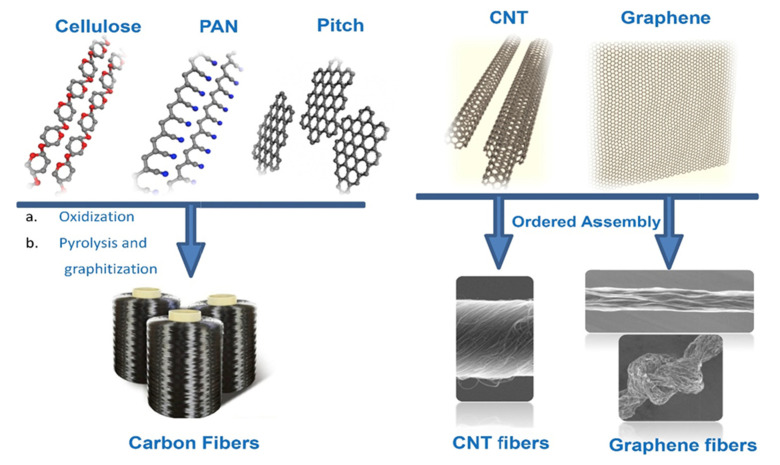
The precursor system of conventional carbon fibers, CNT, and graphene fibers [113].

**Figure 11 polymers-13-03337-f011:**
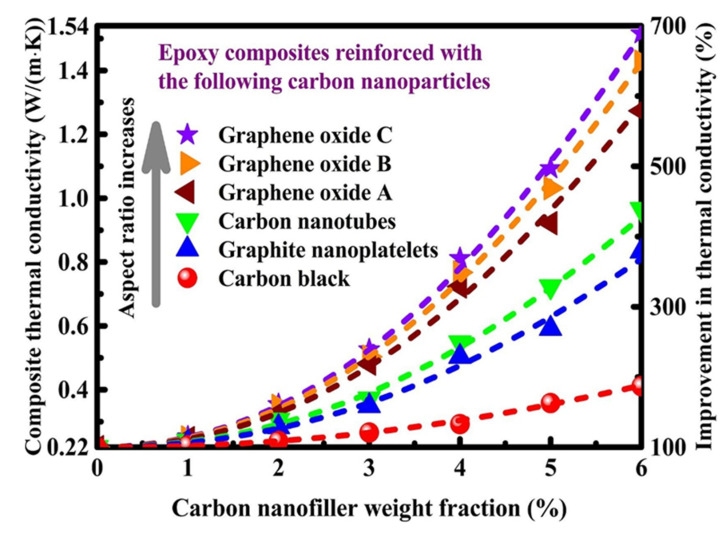
Thermal conductivity of carbon nano filler-based epoxy matrix composite materials at room temperature [129].

**Figure 12 polymers-13-03337-f012:**
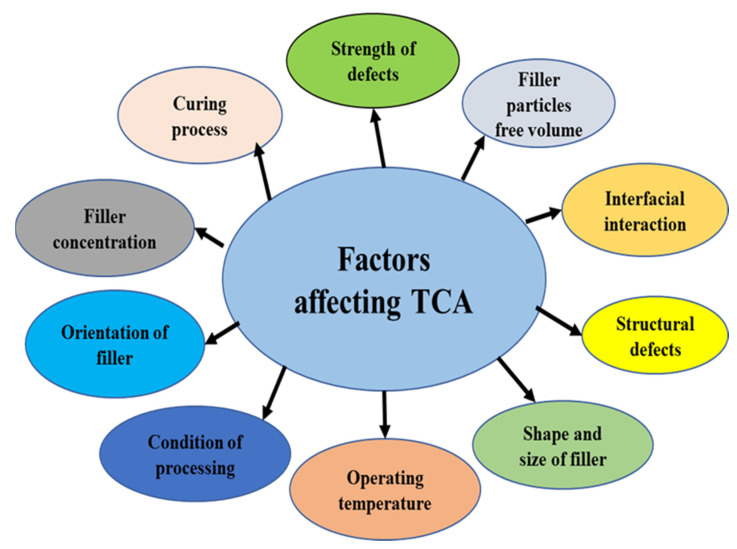
Common factors that can affect a TCA.

**Table 1 polymers-13-03337-t001:** Thermal conductivity (TC) of materials used as conductive fillers [36,37,38,39,40].

Material	TC (Wm^−1^ K^−1^)	Material	TC (Wm^−1^ K^−1^)
Aluminum oxide	20–30	Graphite	100–400 (on plane)
Molybdenum	142	Silver	450
Tungsten	155	Copper	401
Nickel	158	Silicon carbide (SiC)	490
Aluminum	204	Diamond	2000
Beryllium oxide	260	Boron nitride	~2000 (in-plane); ~380 (out-of-plane)
Carbon fiber	260	Multiwalled carbon nanotube (MWCNT)	~3000
Aluminum nitride (AlN)	200–320	Graphene	~5300
Gold	345	Single-walled carbon nanotube (SWCNT)	~6000

**Table 2 polymers-13-03337-t002:** Summary of the recently published works on ceramic fillers.

Filler	Conditions/Process	TC (Wm^−1^ K^−1^)	References
BN	Platelet-shaped Boron Nitride(BN) particles	3.5	[73]
BN	At 70 wt% functionalized and mix with epoxy resin	2.8	[50]
BN	Admicellar-treated BN particles.	2.7	[74]
BN	30 wt% of BN particles modified by 3-aminopropyl triethoxysilane	1.178	[75]
BN	Hexagonal BN/epoxy composites at 44 vol% (densely packed and vertically aligned).	9	[76]
BN	Hexagonal, cubic, and conglomerated -BN.	2.91, 3.95, and 10.1	[77]
BN	Hexagonal boron nitride laminates	20	[78]
BN	Untreated and OTAB-treated BN/epoxy composites.	1.9 and 3.4	[79]
BN	88 wt% of BN loading.	32.5	[80]
AlN	58.4 vol% of large-sized Aluminum nitride (AlN) with small-sized Al_2_O_3_	2.842 and 3.4	[81]
AlN	29 wt% of MWCNTs/AlN	1.04	[82]
AlN	20 vol% AlN particles (magnetically aligned)	1.8	[83]
AlN	50 wt% of 5 μm-AlN particles and 6 wt% of GO	2.77	[65]
AlN	67 vol% of AlN particles (large-sized silane-coated).	14	[84]
AlN	Cycloaliphatic epoxy/trimethacrylate system	0.47	[85]
AlN	At 47 vol% nano-whiskers AlN	4.2	[86]
Al_2_O_3_	At 80 wt% of Alumina (Al_2_O_3_)/epoxy, filled with 5 wt% of graphene oxide (GO) and 5 wt% of Al(OH)_3_-coated GO	3.5 and 3.1	[87]
Al_2_O_3_	Al_2_O_3_/GFRP (amino group grafted)	1.07	[17]
Al_2_O_3_	At 60 vol% of micron-sized alumina	4.3	[88]
SiC	Magnetically aligned BN and Silicon Carbide (SiC) filler system	5.77	[70]
SiC	Nano-sized SiC particles with triethylenetetramine (TETA) functionalized MWCNTs, (at 30% vol%)	2.00	[89]
SiC	At 20 vol% of SiC particles (magnetically aligned Fe_3_O_4_ coated)	1.681	[90]

**Table 3 polymers-13-03337-t003:** Summary of the recently published works on carbon-based fillers.

Filler	Conditions/Process	TC (Wm^−1^ K^−1^)	References
CNT	1–5 vol% BTC-MWCNTs	0.96	[134]
CNT	At 14.8 vol% of CNT (axial and transverse direction)	1.85 and 2.41	[135]
CNT	At 1 wt% of double walled CNT and 0.01 wt% of graphene.	∼12	[35]
CNT	3D CNT reinforced exfoliated graphite block	∼38	[136]
Carbon fiber	At 56 vol% of carbon fiber	291	[137]
Carbon fiber	carbon fiber/epoxy composites	1.329	[138]
Carbon fiber	vapor grown carbon fiber (VGCF)/epoxy	∼695	[137]
Graphene	Aligned MLG/epoxy composite system	33.54	[139]
Graphene	at 6 vol% with epoxy	2.13	[140]
Graphene	GNPs reinforced polymer composites.	12.4	[141]
Graphene	At 20 wt% of GNP with different particle size	1.8 and 7.3	[142]
Graphene	Layer-by-layer assembly of (GO) on a flexible NFC substrate.	12.6	[143]

## Data Availability

Not applicable.

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
