# Peer review of "Recent Advances on Thermally Conductive Adhesive in Electronic Packaging: A Review"

_polymers, 2021, doi:10.3390/polym13193337_

Round 1
Reviewer 1 Report
- As a review article, the references are too old, because the number of literature in the last three years is lacking about thermally conductive adhesive in electronic packaging
- The summary of the literature needs to be further studied, the interaction mechanism of different fillers should be analyzed deeply.
Such as “It is found that the 2D-BN nanoplates filler reduced the mechanical strength, but the thermal conductivity is 30% better than the without filler sample.” Why?
- “The commonly carbon-based fillers are graphite, carbon nanotubes, reduced graphene, graphene oxide, carbon black, and carbon fiber.” It is better to give examples on their applications in thermally conductive adhesive of electronic packaging.
- “Using low-cost substrate materials and high-thermally conductive filler materials should be prioritized. Thermal management of electronic devices is becoming easy by developing the TCAs, but we still have the challenges to produce the TCAs”
what are the detailed challenges? And the solutions?
Author Response
The authors would like to thank you and all the reviewers for their detailed and constructive comments to further strengthen this manuscript. We are extremely grateful to all our reviewers for the effort and precious time put into reviewing this manuscript. Each comment has been carefully considered and responded. The corresponding changes made in the revised manuscript are summarized below. The comments from all reviewers are in normal font and our responses are italicized.
Reviewer-1:
- As a review article, the references are too old, because the number of literatures in the last three years is lacking about thermally conductive adhesive in electronic packaging.
Response: The authors agreed with the comment made by the reviewer. According to the reviewer’s suggestion, most recently published articles on thermally conductive adhesive in electronic packaging have been cited. In this revised form of manuscript, there are 93 references from the last five years among total 149 references. The author would like to thank the reviewer for the comment.
- The summary of the literature needs to be further studied; the interaction mechanism of different fillers should be analyzed deeply.
Such as “It is found that the 2D-BN nanoplates filler reduced the mechanical strength, but the thermal conductivity is 30% better than the without filler sample.” Why?
Response: The author would like to thank the reviewer for the comment. The interaction mechanism of different fillers has been revised and more details has been added in section 3. The mentioned statement on BN has been improved in revised manuscript (page 8) and highlighted in yellow as- “BN nanoplates filler affects thermal conductivity of the composite with the increase in filler concentrations. It is found that the 2D-BN nanoplates filler at 14 wt.% with silicone can reduce the mechanical strength, but the thermal conductivity is 30% better than the without filler sample. The reduction in the mechanical strength can be due to agglomeration of BN nanoparticles, whereas the increase in the thermal conductivity is due to the acceleration of phonons transmission by the BN nanoplates”.
- The commonly carbon-based fillers are graphite, carbon nanotubes, reduced graphene, graphene oxide, carbon black, and carbon fiber.” It is better to give examples on their applications in thermally conductive adhesive of electronic packaging.
Response: The author would like to thank the reviewer for the suggestion. The statement regarding the application of commonly carbon-based filler in thermally conductive adhesive has been revised and supported by references (Ref No 108 to 111). The changes have been highlighted in the revised manuscript (page-13).
- Using low-cost substrate materials and high-thermally conductive filler materials should be prioritized. Thermal management of electronic devices is becoming easy by developing the TCAs, but we still have the challenges to produce the TCAs” what are the detailed challenges? And the solutions?
Response: The authors would like to thank the reviewer for the comment. The challenges and solutions to produce the TCAs have been stated in the revised manuscripts (page 17) as- “Based on this extensive review of the recent research, it is found that the shape of the filler is a critical factor but often overlooked in the thermal conductivity improvement of TCAs. Other big challenges to progress in the TCA sector include developing low-dimensional materials with a high aspect ratio, dispersing them into the matrix, achieving high thermal conductivity while being electrically insulating, and developing a novel heat conduction network. Undoubtedly, high thermal conductivity for TCA is one of the biggest challenges, but the following research potentials are for improving thermal conductivity of TCA’s:
(1) The improvement of nanomaterial preparation techniques and process parameters can contribute to the development of an efficient three-dimensional thermal conductive network in matrix.
(2) Surface modification seems to be an effective method of reducing the thermal resistance at the interface. however, it leads to reduction in the low-dimensional materials' intrinsic thermal conductivity.
(3) Metallic and carbon based (graphene and carbon nanotubes) fillers have high thermal conductivity but possess high electron mobility. The ceramic fillers (BN, AlN, SiC, and Al2O3) are highly thermally conductive but electrically insulated. So, the hybridization of filler can be a new research direction.
(4) The wonder material-graphene oxide (GO) appears to be a potential choice because of its solution processability and controllable deposition on the substrate. GO has been used on various substrates, but there is still room for development in terms of adhesion and heat transmission.
Reviewer 2 Report
In the manuscript authors present a review summarizing the current state of the science in the field of thermally conductive adhesive in electronic packaging. The topic is interesting and prospective. However, the manuscript can be recommended for publication in Polymers journal only after rewriting of some sections. The two main issues that should be addressed by authors are the following:
1. Literature references cited in this review are not up to date. The introduction is one of the main sections of a review article. But in this section the state-of-art in the field of thermally conductive adhesive in electronic is not presented properly. Only 2 articles are referenced covering the period later then 2019. Additionally, in introduction authors mention that “Many review articles also have been published”, but no references are given. This should be fixed. Thus, introduction section and all manuscript should be further supplemented and reviewed again before considering for publication.
2. Conclusions of the review should clearly indicate future perspectives and gaps in the field that needs further investigation. In the current state this is not clear from the text.
Other minor remarks are:
1. Columns 2 in Tables 2 and 3 are difficult to read. Please consider reformatting this Tables.
2. Change the style of the bibliography references according to the requirements of the journal.
3. Please check all manuscript for typos. A lot of extra and missing spaces. It is necessary to check the whole text of the manuscript.
For example:
Line 84: Fig.-2. and so on.
Line 219: 106 %
Line 322: Wm-K[104].
W/m-K should be W·m -1·K-1
Author Response
The authors would like to thank you and all the reviewers for their detailed and constructive comments to further strengthen this manuscript. We are extremely grateful to all our reviewers for the effort and precious time put into reviewing this manuscript. Each comment has been carefully considered and responded. The corresponding changes made in the revised manuscript are summarized below. The comments from all reviewers are in normal font and our responses are italicized.
Reviewer-2:
In the manuscript authors present a review summarizing the current state of the science in the field of thermally conductive adhesive in electronic packaging. The topic is interesting and prospective. However, the manuscript can be recommended for publication in Polymers journal only after rewriting of some sections. The two main issues that should be addressed by authors are the following:
- Literature references cited in this review are not up to date. The introduction is one of the main sections of a review article. But in this section the state-of-art in the field of thermally conductive adhesive in electronic is not presented properly. Only 2 articles are referenced covering the period later then 2019. Additionally, in introduction authors mention that “Many review articles also have been published”, but no references are given. This should be fixed. Thus, introduction section and all manuscript should be further supplemented and reviewed again before considering for publication.
Response: The authors agreed with the comment made by the reviewer. The introduction section has been improved, and references are taken from the latest published articles. The mentioned statement (page 2) has been revised and supported by recent published review articles. Other sections of the manuscript also have been improved and highlighted in the revised manuscript. The authors would like to thank the reviewer for the comment.
- Conclusions of the review should clearly indicate future perspectives and gaps in the field that needs further investigation. In the current state this is not clear from the text.
Response: The authors would like to thank the reviewer for this valuable comment. The conclusion section has been improved. Future perspectives and gaps are important for further investigation in the research field. To comply with the reviewer-3, this topic has been discussed and highlighted in the Challenges and research potential section(page-17).
Other minor remarks are:
- Columns 2 in Tables 2 and 3 are difficult to read. Please consider reformatting this Tables.
- Change the style of the bibliography references according to the requirements of the journal.
- Please check all manuscript for typos. A lot of extra and missing spaces. It is necessary to check the whole text of the manuscript.
For example:
Line 84: Fig.-2. and so on.
Line 219: 106 %
Line 322: Wm-K[104]. W/m-K should be W·m -1·K-1
Response: The authors would like to thank the reviewer for the comment. Columns 2 in Table 2 and 3 have been modified, and now it has become easier to read. The bibliography style of the references has been revised according to the journal requirement. The authors would like to apologize for the typos. Thorough checking has been carried out and, errors have been eliminated in the revised manuscript. The manuscript is further checked using Grammar Software.
Reviewer 3 Report
General comments: This study comprising the fundamentals of thermally conductive adhesive (TCA), formation, and heat transfer mechanism, the recent advances of TCA by ceramic, metallic, and carbon-based fillers and TCA applications in thermal management of electronic packaging.
Specific Issues :
- Please eliminate multiple references. After that, please check the manuscript thoroughly and eliminate ALL the lumps in the manuscript. This should be done by characterising each reference individually and by mentioning 1 or 2 phrases per reference to show how it is different from the others and why it deserves mentioning. Multiple references are of no use for a reader and can substitute even a kind of plagiarism, as sometimes authors are using them without proper studies of all references used. In the case, each reference should be justified by it is used and at least short assessment provided.
2. Too many non-content words may indicate wordiness. Consider rewriting to avoid some of these words: the, for, of, into, via, as well as, their, with, are, also, which. For the text clarity would you refrain from using additional words, mostly meaningless filler words, which can be omitted or some archaic words see e.g. "respectively", "thus", "hence", therefore", "furthermore", "thereby", "basically,", "meanwhile"," wherein", "herein", "hitherto", "Nonetheless", "Perceivably" , "whereas",etc. ?
3. In section two (2), clear explain the heat transfer mechanism by adding numerical equations and basic mechanism.
4. Figure 3 is very basic because it is a detailed review. Authors should use figures to explain the formulation in detail.
5. Figure 5 reference?, similarly other Figures need to be checked?
6. Critical writeup and justification were missing in this paper. The authors need to rewrite Section 3. For Eg " The results demonstrate that large particles are more conducive to the heat conductivity of epoxy adhesive than small particles (shown in Fig.-7)". However, the author did not relate in terms of the critical outcomes of this research and the remaining research gaps.
7. Section 3.3, needs a more detailed analysis especially in the CNT section and other sections too.
8. Before Section 4 , the Authors need to add a new section Economic Perspective of the current work.
9. Section 4 is very weak and need to rewrite completely, this is a very important section and need to be divided into two parts. Part 1 with heading challenges covers environmental, physical, theoretical, material, and other challenges. Part 2 should be Future work, in which the Author need to write a research gap which they learn by doing a critical literature review in the above sections.
Author Response
The authors would like to thank you and all the reviewers for their detailed and constructive comments to further strengthen this manuscript. We are extremely grateful to all our reviewers for the effort and precious time put into reviewing this manuscript. Each comment has been carefully considered and responded. The corresponding changes made in the revised manuscript are summarized below. The comments from all reviewers are in normal font and our responses are italicized.
Reviewer-3:
General comments: This study comprising the fundamentals of thermally conductive adhesive (TCA), formation, and heat transfer mechanism, the recent advances of TCA by ceramic, metallic, and carbon-based fillers, and TCA applications in thermal management of electronic packaging.
Specific Issues:
- Please eliminate multiple references. After that, please check the manuscript thoroughly and eliminate ALL the lumps in the manuscript. This should be done by characterising each reference individually and by mentioning 1 or 2 phrases per reference to show how it is different from the others and why it deserves mentioning. Multiple references are of no use for a reader and can substitute even a kind of plagiarism, as sometimes authors are using them without proper studies of all references used. In the case, each reference should be justified by it is used and at least short assessment provided.
Response: The authors are fully agreed with the comment made by the reviewer. Thorough checking on the references has been carried out in the revised manuscript. Multiple references have been separated and cited as single reference with the relevant phrases. The authors would like to thank the reviewer for this valuable suggestion.
Too many non-content words may indicate wordiness. Consider rewriting to avoid some of these words: the, for, of, into, via, as well as, their, with, are, also, which. For the text clarity would you refrain from using additional words, mostly meaningless filler words, which can be omitted or some archaic words see e.g. "respectively", "thus", "hence", therefore", "furthermore", "thereby", "basically,", "meanwhile"," wherein", "herein", "hitherto", "Nonetheless", "Perceivably" , "whereas",etc. ?
Response: The authors would like to thank the reviewer for the comment. Thorough checking on the references has been carried out and, errors have been eliminated in the revised manuscript.
- In section two (2), clear explain the heat transfer mechanism by adding numerical equations and basic mechanism.
Response: Heat transfer mechanism in section-2 (page 5 and 6)) has been improved by adding numerical equations and describing basic mechanism. The improved portion has been highlighted in the revised manuscript. The authors would like to thank the reviewer for the comment.
- Figure 3 is very basic because it is a detailed review. Authors should use figures to explain the formulation in detail.
Response: The authors agreed with the comment made by the reviewer. Figure 3, explaining the formulation process of TCA has been included in the revised manuscript (page 4). The authors would like to thank the reviewer for the comment.
- Figure 5 reference?, similarly other Figures need to be checked?
Response: Figure no 2, 4, 5, 8, and 12 in the manuscript have been made by the authors, so reference not required. Figure no 1, 3, 6, 7, 9, 10, and 11 have been cited after receiving permission of reuse from the publisher. The author would like to thank the reviewer for the comment.
- Critical writeup and justification were missing in this paper. The authors need to rewrite Section 3. For Eg " The results demonstrate that large particles are more conducive to the heat conductivity of epoxy adhesive than small particles (shown in Fig.-7)". However, the author did not relate in terms of the critical outcomes of this research and the remaining research gaps.
Response: The author would like to thank the reviewer for the comment. The mentioned statement has been revised(page-9) and highlighted as- “The thermal conductivity of adhesive is mainly determined by the filler heat transfer capacity, density of thermal network, as well interfacial thermal resistance. Thus, the formation of effective thermal flow 3-D percolating network through synergistic effect in matrix is a crucial criterion, dominating the thermal conductivity. Yuan et. al(reference-) worked on different sized (5 µm, 2 µm and, 50 nm) AlN- with graphite and graphene oxide (GO) as hybrid filler to observe the improvement of thermal conductivity of composite. The results demonstrate that large particles of AlN with epoxy are more heat conductive than small particles. Similarly, GO can improve the thermal conductivity of epoxy resin more effectively than natural graphite. In the case of a single filler, adding 70 wt% 5 µm-AlN particles to the epoxy resulted in the maximum conductivity which is 10.8 times that of pure epoxy (shown in Figure 7). The whole section also has been improved according to the suggestion and highlighted in the revised manuscript. The outcome of the research and remaining research gaps have been discussed in the challenges and research potential (section 5).
- Section 3.3, needs a more detailed analysis especially in the CNT section and other sections too.
Response: Section 3.3 has been improved according to the reviewer suggestion. Other subsections of carbon-based fillers including CNT have been revised and highlighted in the revised manuscript. The author would like to thank the reviewer for this valuable comment.
- Before Section 4, the Authors need to add a new section Economic Perspective of the current work.
Response: The authors agreed with the comment made by the reviewer. Economic perspective of TCA (section 4) has been included before the Challenges and research potential section. This section describes the world market of TCA and the recently published works on cost effective conductive filler production.
- Section 4 is very weak and need to rewrite completely, this is a very important section and need to be divided into two parts. Part 1 with heading challenges covers environmental, physical, theoretical, material, and other challenges. Part 2 should be Future work, in which the Author need to write a research gap which they learn by doing a critical literature review in the above sections.
Response: According to the reviewer suggestion, the Challenges and research potential section has been rewritten. This section has been divided into two parts. First part describes the challenges, and the second part describes the research potential to overcome the challenges. The improved portions have been highlighted in the revised manuscript. The authors would like to thank the reviewer for the comment.
Round 2
Reviewer 1 Report
That's ok for answering the questions, the revision is good.Reviewer 2 Report
All comments have been addressed.
Reviewer 3 Report
All comments are successfully addressed.